# Hypercharge quantisation and Fermat's last theorem

**Nakarin Lohitsiri[1] and David Tong[1,2]**

**1** Department of Applied Mathematics and Theoretical Physics,
University of Cambridge, Cambridge, CB3 OWA, UK
**2** School of Physics, Korea Institute for Advanced Study
85 Hoegi-ro Dongdaemun-gu, Seoul 02455, Korea

## Abstract

What values of the Standard Model hypercharges result in a mathematically consistent quantum field theory? We show that the constraints imposed by the lack of gauge anomalies can be recast as the equation $x^3 + y^3 = z^3$. If hypercharge is quantised, then $x$, $y$ and $z$ must be integers. The trivial (and only) solutions, with $x = 0$ or $y = 0$, reproduce the hypercharge assignments seen in Nature. This argument does not rely on the mixed gauge-gravitational anomaly, which is automatically vanishing if hypercharge is quantised and the gauge anomalies vanish.

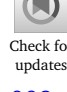
The delicate cancellation of gauge and mixed gauge-gravitational anomalies reveals the Standard Model to be a wonderfully elegant jigsaw, each piece interlocking perfectly with the others [1–3]. One could ask: is there another way to put the pieces together? In particular, are there other assignments of hypercharge that would also result in a consistent theory?

There are different ways of posing this question. For example, we could take the gauge group of the Standard Model to be,

$$G = \mathbb{R} \times SU(2) \times SU(3).$$

Here the unfamiliar factor of $\mathbb{R}$ reflects the fact that we do not impose any quantisation condition on the hypercharge. We take a single generation of fermions, sitting in the usual representations of the non-Abelian part of the gauge group, but with arbitrary hypercharges,

$$q_L : (\mathbf{2}, \mathbf{3})_q, \quad l_L : (\mathbf{2}, \mathbf{1})_l, \quad u_R : (\mathbf{1}, \mathbf{3})_u, \quad d_R : (\mathbf{1}, \mathbf{3})_d, \quad e_R : (\mathbf{1}, \mathbf{1})_x.$$

The resulting quantum field theory is consistent only if the hypercharges $\{q, l, u, d, x\}$, each of which is a real number, are constrained to obey three anomaly conditions. Two of these are linear, arising from the vanishing of the mixed anomalies between Abelian and non-Abelian gauge groups

$$2q - u - d = 0 \quad \text{and} \quad 3q + l = 0. \tag{1}$$

The third is a cubic equation arising from the Abelian triangle anomaly,

$$6q^3 + 2l^3 - 3u^3 - 3d^3 - x^3 = 0. \tag{2}$$

There are an infinite number of solutions to these equations with hypercharges valued in $\mathbb{R}$. In particular, there are an infinite number of solutions with $x/q$ irrational. This means that if we do not impose quantisation of charge then the gauge anomaly constraints do not impose it for us.

In addition, we could quite reasonably ask that the Standard Model can be consistently coupled to gravity. This gives a further linear constraint, arising from the mixed gauge-gravitational anomaly [4–6],

$$6q + 2l - 3(u + d) - x = 0\,. \tag{3}$$

It is well known that there are two solutions to these anomaly equations, [7–10]. The first solution is somewhat trivial,

$$q = l = x = 0 \ \text{ and } \ u = -d\,. \tag{4}$$

The second is, up to an overall rescaling, the charge assignment seen in Nature,

$$x = 2l = -3(u + d) = -6q \ \text{ and } \ u - d = \pm 6q\,. \tag{5}$$

Both solutions result in a quantised hypercharge, in the sense that the ratios of all charges are rational. This means that the joint requirements of gauge and gravitational consistency imply charge quantisation, even though this wasn't imposed from the outset.

In this paper, we show the converse: charge quantisation, together with vanishing gauge anomalies, is sufficient to ensure cancellation of the gravitational anomaly. To this end, we take the gauge group of the Standard Model to be (omitting possible discrete quotients)

$$G = U(1) \times SU(2) \times SU(3)\,,$$

with the $U(1)$ factor normalised so that all charges are integers. We now wish to find integer solutions to the gauge anomaly conditions (1) and (2). Such Diophantine equations are, in general, hard to solve. Recently, a number of methods have been developed to find integer solutions to the anomaly constraints in different quantum field theories [11–17]. For the Standard Model, with a single generation, it turns out that there is a remarkably quick way to find all solutions.

We will show that there are precisely two integer solutions to (1) and (2), namely (4) and (5). Each of these solutions automatically satisfies the mixed gauge-gravitational anomaly condition (3). In other words, insisting on a $U(1)$ gauge group, rather than $\mathbb{R}$, is sufficient to ensure consistency with gravity.

This statement is a little surprising. It is certainly not true that general chiral gauge theories with a $U(1)$ factor can be coupled to gravity. Indeed, the first consistent 4d chiral gauge theory was constructed by Ramanujan from his hospital bed in Putney and suffers a mixed gauge-gravitational anomaly [1].

To prove the claim, note that the first equation in (1) tells us that the sum of hypercharges $u + d$ is even. Therefore the difference is also even and we can write $u - d = 2y$. Using the second equation in (1) to set $l = -3q$, the remaining cubic equation (2) becomes

$$x^3 + 18qy^2 + 54q^3 = 0\,. \tag{6}$$

Our goal is to find integer solutions to this equation. There is the trivial solution with $x = q = 0$; this corresponds to (4). Any further solution necessarily has $q \neq 0$. Because (6) is a homogeneous polynomial we may, without loss of generality, rescale to set $q = 1$ and look for rational solutions to the curve

$$x^3 + 18y^2 + 54 = 0 \quad x, y \in \mathbb{Q}\,. \tag{7}$$

---

[1] $1729 = 1^3 + 12^3 = 9^3 + 10^3$. Ramanujan also constructed a two parameter family of integer solutions to $x^3 + y^3 + z^3 = w^3$. This is described on page 158 of [14].

This is a rather special elliptic curve. To see this, we introduce two new coordinates $v, w \in \mathbb{Q}$, defined by

$$x = -\frac{6}{v+w} \quad , \quad y = \frac{3(v-w)}{v+w} \, .$$

This reveals the elliptic curve (7) to be the Fermat curve

$$v^3 + w^3 = 1 \, . \tag{8}$$

Any non-trivial rational solution to this equation would imply a non-trivial integer solution to the equation $v^3 + w^3 = z^3$. There are none [18]. The trivial solutions to (8) are $v = 1, w = 0$ and $v = 0, w = 1$. These reproduce the hypercharge assignments (5) of the Standard Model.

We could also repeat the story above with a right-handed neutrino. With a Majorana mass, the right-handed neutrino is forbidden from carrying hypercharge and the results above are unchanged. In the absence of a Majorana mass, things are not so pretty. We ascribe hypercharge $n$ to the right-handed neutrino. With gauge group $\mathbb{R}$, it is simple to check that the combined gauge and gravitational anomalies no longer impose quantisation of charges. If, instead, we insist on gauge group $U(1)$ then equation (8) is replaced by the Fermat surface

$$v^3 + w^3 + t^3 = 1 \, ,$$

where $n = 6t/(v+w)$. Now there are many non-trivial rational solutions, including the taxi-cab numbers. However, in this case cancellation of the mixed gauge-gravitational anomaly occurs only for the trivial solutions in which two of the numbers coincide. These solutions are given as a 2-parameter family of rational linear combination by the Standard Model hypercharge and $\mathrm{B} - \mathrm{L}$.

# Acknowledgements

We thank Joe Davighi, Bogdan Dobrescu, Paddy Fox, Jerome Gauntlett and Scott Melville for comments. We also thank 79 Facebook friends for confirming that they were previously unaware of this result. DT is grateful to KIAS for their kind hospitality while this work was done. We are supported by the STFC consolidated grant ST/P000681/1. DT is a Wolfson Royal Society Research Merit Award holder and is supported by a Simons Investigator Award. NL is supported by the Internal Graduate Scholarship from Trinity College, Cambridge, and by the Thai Government.

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
