# Peer review of "Hypercharge Quantisation and Fermat's Last Theorem"

_SciPost Physics, doi:SciPost Phys. 8, 009 (2020)_

## Round 1 · Referee Report · Joseph Minahan (Referee 1) · 2019-8-12

Report

In this paper the authors look at the problem of cancelling anomalies in the Standard Model. A long time ago it was realized that by including the cancellation of the mixed gauge-gravitational anomalies along with the usual gauge anomalies in the Standard Model, then there are two possible solutions for the hypercharges [4,3], one of which is the solution found in nature. Both solutions are rational.

The authors here assume that the hypercharge gauge symmetry is $U(1)$ and not $\mathbb{R}$ and so the rationality of the hypercharges is an initial condition. They drop the requirement that the mixed anomalies cancel and instead only assume cancellation of the gauge anomalies. After a change of variables they are able to turn the anomaly equation for the hypercharges into Fermat's cubic equation, which was proven by Euler to only have trivial solutions in the rationals. Those solutions correspond to the same ones found in [4] and [3]. Very nice.

Requested changes

The requested changes only concern references.

  1. A reference to Georgi and Glashow, Phys.Rev. D6 (1972) 429 should be included in [1]. While not explicitly spelled out in the paper, the choice of the spectrum in the second unnumbered equation line has no $SU(3)^3$ anomalies.
  2. The reference for the mixed gauge-gravity anomaly in [2] should also include Delbourgo and Salam, Phys.Lett. 40B (1972) 381-382 and Eguchi and Freund, Phys.Rev.Lett. 37 (1976) 1251
  3. Since [4] preceded [3] by 7 years, shouldn't that reference go first? It is also hard to say how well known the result was at the time. Is there any reference before [4] discussing this?

  • validity: -
  • significance: -
  • originality: -
  • clarity: -
  • formatting: -
  • grammar: -

Author:  David Tong  on 2019-09-13  [id 604]

(in reply to Report 1 by Joseph Minahan on 2019-08-12)

Dear Joe,

Thanks very much for this report and for the suggested referencing.

Nakarin and David

---

## Round 1 · Referee Report · Isabel García García (Referee 2) · 2019-8-24

Strengths

In this paper, the authors try to gain some insight into the potential interplay between the hypercharge assignments seen in nature and theoretical consistency in the framework of a gravitational theory. After reviewing the known result that cancellation of gauge and gravitational anomalies implies charge quantization, the authors show that, in turn, requiring absence of gauge anomalies plus charge quantization implies cancellation of the gravitational anomaly. This is a noble endeavour, particularly in light of the common lore that in any consistent quantum gravity theory all gauge groups must be compact. The authors' result, although subject to certain assumptions, appears to be correct.

Weaknesses

That said, I believe that one of those assumptions is limiting enough that, before publication, the authors should either address it, or state it more clearly. By this I mean omission of the fact that right-handed (RH) neutrinos could, in principle, carry non-zero hypercharge. Given that neutrino masses have been measured to be non-vanishing, I think any attempt to understand the pattern of hypercharge assignments seen in nature from some kind of first principles should at least acknowledge this.

Report

Two comments:

(i) As I am sure the authors already know, the previous result already present in the literature no longer holds if one allows for the hypercharge of $\nu_R$ to be $n \neq 0$. Demanding that the theory be free of both gauge and gravitational anomalies, there are two possible hypercharge assingments. The first, with $q=0$, is the generalization of Eq.(4) in [1]:
\begin{equation}
q = l = 0 \ , \qquad u = - d = \gamma \ , \qquad n = - x = \delta \ ,
\end{equation}
where $\gamma , \delta \in \mathbb{R}$. In particular, the ratio $\gamma / \delta$ could be taken to be an irrational number. The second, with $q \neq 0$, generalizes Eq.(5) in [1], and reads:
\begin{equation}
q = - \frac{l}{3} = \frac{\alpha}{6} + \frac{\beta}{3} , \ u = \frac{2 \alpha}{3} + \frac{\beta}{3} , \ d = - \frac{\alpha}{3} + \frac{\beta}{3} , \ x = - \alpha - \beta , \ n = - \beta ,
\end{equation}
where $\alpha , \beta \in \mathbb{R}$. For example, by taking $\alpha = 1$ and $\beta$ an irrational number, some of the charge ratios will be irrational. I think this (known) fact should be acknowledged as a caveat in the paper.

(One could further ask the requirement that hypercharge assignments are such that non-zero fermion masses can be written in the Standard Model (that is, with a single Higgs doublet, and through renormalizable operators). For the $q=0$ solution this would only be possible if $\gamma = \delta = h$ (with $h$ the hypercharge of $H$), and so that would result in quantized charge after all. For the $q \neq 0$ solution, however, that's not true: one can have masses for all fermions so long as $h = \alpha / 2$, while still taking $\beta$ irrational.)

(ii) It is not so clear to me whether the converse statement (i.e. what the authors prove) is similarly modified or not in the presence of RH neutrinos. In this case, Eq.(6) in [1] would now read
\begin{equation} n^3 + x^3 + 18 y^2 q + 54 q^3 = 0 \ .
\end{equation}
The question would then be whether there are integer solutions to this equation, and to Eq.(1) in [1], that are *not* solutions to the gravitational anomaly equation, which can be rewritten as:
\begin{equation} 6q + x + n = 0 \ . \end{equation}
For $q = 0$, it is clear that to satisfy the $U(1)^3$ anomaly condition one needs $x = - n$, in which case the gravitational anomaly also cancels. For $q \neq 0$, I don't know of any examples in which all gauge anomalies cancels but the mixed gravitational anomaly doesn't. It may be possible to prove this (or show that it is false!), and I believe it would make the paper more relevant if the authors addressed this point.

Requested changes

See last sentence in points (i) and (ii) of the main report.

  • validity: -
  • significance: -
  • originality: -
  • clarity: -
  • formatting: -
  • grammar: -

Author:  David Tong  on 2020-01-16  [id 710]

(in reply to Report 2 by Isabel García García on 2019-08-24)

Dear Isabel,

Thank your very much for your thoughtful and detailed report. And please accept our apology for not getting back sooner. One of the authors (DT) totally dropped the ball on this!

We completely agree with your comments on the RH neutrino. Of course, if there is a Majorana mass there can be no hypercharge for this object (although this may, admittedly, be putting the cart before the horse). If we allow a hypercharge, it's no longer true that gauge anomaly cancellation automatically implies the gauge-gravity anomalies vanish. We've added some comments along these lines to the draft.

Thanks again,

David and Nakarin

---

## Round 3 · Author Response

We thank the referees for their comments. And we apologise for the long delay in getting this updated.

---

## Round 3 · List of Changes

Added several new references as requested in the first report.
Added an extra, final paragraph discussing the right-handed neutrino, as requested in the second report.
Added an extra, final paragraph discussing the right-handed neutrino, as requested in the second report.

---

## Editorial Decision

published